# Recent Advances in the Rheumatic Fever and Rheumatic Heart Disease Continuum

**DOI:** 10.3390/pathogens11020179

**Published:** 2022-01-28

**Authors:** Joselyn Rwebembera, Bruno Ramos Nascimento, Neema W. Minja, Sarah de Loizaga, Twalib Aliku, Luiza Pereira Afonso dos Santos, Bruno Fernandes Galdino, Luiza Silame Corte, Vicente Rezende Silva, Andrew Young Chang, Walderez Ornelas Dutra, Maria Carmo Pereira Nunes, Andrea Zawacki Beaton

**Affiliations:** 1Department of Adult Cardiology (JR), Uganda Heart Institute, Kampala 37392, Uganda; 2Departamento de Clinica Medica, Faculdade de Medicina da Universidade Federal de Minas Gerais, Belo Horizonte 30130-100, MG, Brazil; ramosnas@ufmg.br (B.R.N.); 1180.000170@cienciasmedicasmg.edu.br (L.P.A.d.S.); plxbruno@ufmg.br (B.F.G.); luizasilamecorte@hotmail.com (L.S.C.); vicenterezende@ufmg.br (V.R.S.); mcarmo@waymail.com.br (M.C.P.N.); 3Servico de Cardiologia e Cirurgia Cardiovascular e Centro de Telessaude, Hospital das Clinicas da Universidade Federal de Minas Gerais, Avenida Professor Alfredo Balena 110, 1st Floor, Belo Horizonte 30130-100, MG, Brazil; 4Rheumatic Heart Disease Research Collaborative in Uganda, Uganda Heart Institute, Kampala 37392, Uganda; neemaminja@gmail.com; 5School of Medicine, University of Cincinnati, Cincinnati, OH 45229, USA; Sarah.deLoizagaCarney@cchmc.org (S.d.L.); Andrea.Beaton@cchmc.org (A.Z.B.); 6Department of Paediatric Cardiology (TA), Uganda Heart Institute, Kampala 37392, Uganda; twalib90@gmail.com; 7Department of Epidemiology and Population Health, Stanford University School of Medicine, Stanford, CA 94305, USA; aychang@stanford.edu; 8Laboratory of Cell-Cell Interactions, Institute of Biological Sciences, Department of Morphology, Federal University of Minas Gerais, Belo Horizonte 30130-100, MG, Brazil; waldutra@gmail.com; 9National Institute of Science and Technology in Tropical Diseases (INCT-DT), Salvador 40170-970, BA, Brazil; 10Cincinnati Children’s Hospital Medical Center, The Heart Institute, Cincinnati, OH 45229, USA

**Keywords:** rheumatic fever, rheumatic heart disease, advances

## Abstract

Nearly a century after rheumatic fever (RF) and rheumatic heart disease (RHD) was eradicated from the developed world, the disease remains endemic in many low- and middle-income countries (LMICs), with grim health and socioeconomic impacts. The neglect of RHD which persisted for a semi-centennial was further driven by competing infectious diseases, particularly the human immunodeficiency virus (HIV) pandemic. However, over the last two-decades, slowly at first but with building momentum, there has been a resurgence of interest in RF/RHD. In this narrative review, we present the advances that have been made in the RF/RHD continuum over the past two decades since the re-awakening of interest, with a more concise focus on the last decade’s achievements. Such primary advances include understanding the genetic predisposition to RHD, group A *Streptococcus* (GAS) vaccine development, and improved diagnostic strategies for GAS pharyngitis. Echocardiographic screening for RHD has been a major advance which has unearthed the prevailing high burden of RHD and the recent demonstration of benefit of secondary antibiotic prophylaxis on halting progression of latent RHD is a major step forward. Multiple befitting advances in tertiary management of RHD have also been realized. Finally, we summarize the research gaps and provide illumination on profitable future directions towards global eradication of RHD.

## 1. Rheumatic Fever and Rheumatic Heart Disease: A Historical Background

The history of rheumatic fever (RF) and rheumatic heart disease (RHD) is a mixture of remarkable spurts and frustrating gaps in progress. Much of what we know today about the pathogenesis, diagnosis, prevention, and treatment of RHD originates from now high-income areas in the mid-20th century, where improved socioeconomic standards and access to healthcare boosted by extensive research resulted in a striking reduction of disease burden. However, recent years have seen a reemergence of global RHD research and advocacy, which has largely been grassroots, conducted in low resource settings in populations where RHD remains endemic. Figure 1 provides an illustration of these sequential phases in the RF and RHD arena. On the historical backdrop, we capture the recent progress and highlight the many remaining knowledge and practice gaps.

### 1.1. Decades of Discovery: 1930s–1950s

*Streptococcus* [1], and slightly later *Streptococcus pyogenes* (group A strep, GAS), were discovered during the ‘golden age of bacteriology’ [2] in the late 19th and early 20th century. Concurrently, RF was gaining attention as the leading cause of death among individuals between 5 and 20 years [3,4], with the critical link between the two conditions established by 1889 [5]. By 1944, the diagnosis of RF was standardized by Dr. T. Duckett Jones, (The Jones Criteria [6,7,8]), and treatment remained limited to salicylates and bed rest [4]. By the 1950s, antibiotics became more widely available [9,10] and strong evidence emerged that proper treatment of GAS pharyngitis, both with sulfonamides [4,9] and penicillins [11], could prevent the development of RF [4,12,13,14]. Almost concurrently, the utility of secondary antibiotic prophylaxis to prevent RF recurrence was established [15,16,17], long-acting intramuscular was formulated [18], and every-4-week administration of this compound was found to effectively suppress RF recurrences [16,19,20,21,22]. Thereafter, the use of intramuscular long acting benzathine penicillin G (BPG) at monthly intervals was shown to be superior to daily oral penicillin G in the prevention of streptococcal infections and recurrences of RF [23], establishing a key tenet of RHD management which is still accepted as dogma today.

The early 20th century was also a time of RHD surgical pioneering. The first recorded valvotomy for a patient with RHD occurred at the House of the Good Samaritan (HGS), a RF/RHD specialty hospital in 1923 (Cutler and Levine) [4].

### 1.2. Decades of Dissemination: 1960s–1970s

While improved living conditions in high-resource settings (less crowding and better hygiene) drove down the incidence of RF even before many of the scientific advances described above [24,25,26], targeted advocacy, education, and awareness in the second half of the 20th century greatly accelerated progress. Among the leading advocates of that time was the American Heart Association, an organization that largely coalesced to fight ‘Childhood’s greatest enemy [27]. By the 1970s, the annual incidence of RF in the US had plummeted to 0.6 per 100,000 population in those aged 5 to 19 years [28,29], Similar gains were seen during this time in other high-income countries (much of Europe and Australia), and in some isolated lower income countries (such as Cuba [25] and Costa Rica [30]) which implemented comprehensive RF prevention programs, without the benefit of improved socio-economic conditions at that point in time.

### 1.3. Decades of Stagnation: 1970s–2000s

By the late 20th century, RF and RHD were largely controlled in high-income countries, and new research and funding for research in RF/RHD nearly ceased. However, the waning of interest and research [31] only reflected the waning burden among a fraction of the world’s population, but not where the highest proportion of individuals at social risk lived. During this period, the World Health Organization (WHO) briefly invested in a large-scale program to disseminate knowledge and best practices around the globe [32,33,34,35,36]. However, these efforts were not sustained as the problem of RF/RHD was progressively overshadowed and dwarfed by the epidemics of malaria, tuberculosis, yellow fever and especially—HIV/AIDS. Consequently, despite the persisting burden, and proven interventions known to reduce this burden, RF/RHD was steadily relegated until it became a ‘neglected disease’.

### 1.4. The Re-Awakening: From 2000 to Date

However, over the last two decades, there has been a resurgence of interest in RF/RHD. The increasingly wide-spread availability of echocardiography, in particular portable echocardiography, unmasked the large number of children and adults living undiagnosed with RHD in low-resource settings [37,38,39,40,41], and provided irrefutable evidence of the persistent and disparate global burden. In response, calls to action began to emerge from individuals and research groups [31,42,43] and regional meetings, particularly in sub-Saharan Africa, were convened and produced position papers and guiding statements [44,45,46]. These and other regional efforts were crowned by the 2018 World Health Assembly Resolution to end RHD [47]. In addition, research and research funding have increased, with the exciting application of modern technology and research methodologies to build new knowledge in the RHD cascade of management.

## 2. Advances in Understanding the Global Burden of RHD

The last decade has dramatically changed our understanding of the prevailing burden of RHD, from 15 million estimated cases in 2005 [48] to 40.5 million estimated cases in 2019 [49]. However, despite this global trend, surveys in some individual countries such as India have revealed consistent decline in the burden of RHD before and after 2000, using similar screening methods among school children. This particular decline was associated with improvement in the indicators of socioeconomic state and health-care services. Three main advances have led to the knowledge gain of the global burden of RHD, including (a) screening echocardiography; (b) RHD registries and country-wide administrative health data; and (c) big data sources, namely the Global Burden of Disease repository.

### 2.1. Screening Echocardiography (Echo)

RHD remains largely silent until patients become symptomatic with the advanced form of disease. Echo screening [37], which has been one of the most important advancements in our understanding of the global burden of RHD, has resulted from school, community, and clinical echo screening to define the epidemiological burden. Critically, echo screening birthed the ‘latent RHD’ paradigm—rallied as an opportunity for early intervention to prevent disease progression. Active case finding through echo screening has gained further traction in the past decade particularly following the publication of standardized diagnostic criteria by the World Heart Federation (WHF) in 2012 [50]. The value of echo screening in the control of RHD has been a subject of debate [51] over the recent years through the lenses of criteria for ‘screenable’ diseases [52]. However, we have just witnessed the addition of one critical piece of evidence to justify screening—robust data from a landmark randomized-controlled trial strongly suggest that secondary antibiotic prophylaxis prevents the progression of latent RHD to moderate or severe stages of disease, as compared to no prophylaxis [53].

Further developments toward a more pragmatic uptake of echocardiographic screening in LMICs have included: (a) development of hand-held echo devices which are less expensive, generate smaller and shareable file sizes, and have a smaller profile without reliance on wired electricity [51]; (b) task-shifting of echo image acquisition, from highly trained cardiologists and echocardiographers to non-expert health care workers (HCW) with limited training, allowing for expansion of screening capabilities; (c) supportive cloud-based telemedicine programs for remote echo image interpretation; and finally, (d) expanding technological advances [54,55], with several companies now offering artificial intelligence guidance technology built into hand-held devices, thus empowering novice operators to perform screening echoes even after only minimal training, and promising deep learning approaches for automated flagging of abnormalities in the near future [56]. There is also ongoing effort to improve and simplify the WHF criteria for the diagnosis of latent RHD [57], with additional development of risk scores to predict outcomes in both borderline and definite latent RHD [57].

### 2.2. RHD Registries and Big Data

Recent studies in LMICs, mainly registry-based, have provided some illumination on the dire epidemiology of clinical RHD in these endemic settings [49,58,59]. Data from the hospital-based sub-Saharan Africa Survey of Heart Failure (THESUS-HF) registry study showed that RHD was the second major cause of acute heart failure hospitalizations, with an in-patient mortality of 4.2% and 180-day mortality of 17.8% [60]. The VALVAFRIC [61], REMEDY [62], and some national [63,64] registry studies provide the most recent descriptions of chronic RHD complications in patients residing in low- and middle-income countries (LMICs). In summary, clinical disease in LMICs is characterized by late presentation with already established complications [63,64]. The clinical course punctuated by recurrent hospitalizations, and ultimately, unacceptably early mortality (mean age at death of 29.4 years), with most of these deaths occurring within the first 3 months of diagnosis. Similar trends have been described in India and the Northern territory of Australia. The public health and other multi-faceted global impact of RHD has been recently reviewed.

On the global scene, the burden of RHD continues to reveal worldwide disparities with low-income countries bearing the brunt of the disease. RHD currently affects 40.5 million people across the globe and accounts for 306,000 deaths annually—representing 1.6% of all mortality from cardiovascular disease, the leading cause of death worldwide [49]. The increased prevalence is most likely related to improved case detection from increasing availability of echocardiography, better survival and the chronic nature of the disease [49]. Sub-Saharan Africa retains the highest prevalence of clinically apparent RHD [65], and there is some evidence that the true burden of RHD in some African settings may be much higher than previous estimates [66]. Today, the highest age-standardized DALY rates lost to RHD are observed in the regions of Oceania (627.4 per 100,000) and South Asia (348.5 per 100,000). In Australia and New Zealand, the highest rates of RF and RHD are found among indigenous populations [67,68]. Unfortunately, epidemiological data on RHD prevalence is least robust from the most affected regions [69]. National, regional and global RHD registries continue to report female predominance with several postulated theories, however with no gender-based differences in complications such as heart failure, atrial fibrillation, stroke and pulmonary hypertension.

## 3. Advances in Understanding RF/RHD Pathogenesis

The consensus remains that immune-mediated pathology is the hallmark of RHD. Although we have a more refined understanding of the factors contributing to RHD pathogenesis, the exact mechanisms underlying this complex process have yet to be completely unveiled. Traditional dogma holds that GAS pharyngitis is responsible for triggering the immune reaction that leads to RHD pathology [70]. However, more recently, other superficial streptococcal infections, such as impetigo and pyoderma, have been associated with RHD [71,72]. There is also emerging evidence potentially implicating group C and group G streptococcus in contributing to the pathogenesis of RHD [73,74,75,76,77].

The most widely accepted hypothesis in the pathogenesis of RF/RHD [78,79] is molecular mimicry, but more recent data suggests a more complex cascade of events, such as the hypothesis of collagen-associated neo-antigens [80,81] and additional mechanisms such as epitope spreading [82] and T cell receptor (TCR) degeneracy [83,84,85] that may participate in pathology. Additionally, specific CD4+ T cells, as well as NK cells and CD4-CD8- T cells, might be important players in tissue destruction in RHD [86]. Recent efforts have been developed to take advantage of the intense response triggered by GAS antigens toward the establishment of vaccines, which have remained elusive largely due to lack of investment [87]. Dissecting the immunopathogenic responses using state-of-the-art methods such as single cell RNA sequencing combined with TCR CDR3 region usage may open new insights to advance this field.

## 4. Advances in Understanding the Genetic Predisposition to RF/RHD

Only a minority (roughly 3–6%) of GAS-endemic populations develop RF [70] and RHD is largely understood to have complex genetic risk [88]. Historical RHD studies in twins have supported this postulation [89,90], and more recently modern genetic interrogation is shedding new light on RHD pathogenesis and predisposition.

Genetic associations in RHD have been explored through numerous candidate gene studies—reviewed by Martin et al. [91] and Muhamed et al. [92] eliciting genetic associations in RHD through candidate gene studies has yielded conflicting and heterogeneous results, implicating a variety of genes with several listed study limitations [89,91,92] several studies have reported on the association of human leukocyte antigen (HLA) molecules, encoded by genes on chromosome 6, and susceptibility to develop RF/RHD [89,93,94,95,96,97]. Within HLA genes, class II has been widely reported, the majority linking to HLA-DR7 [89,96] and HLA-DR4 [91,95,98].

There has also been interest in evaluating genetic polymorphisms coding for inflammatory mediators in RHD and their phenotypic expression. Polymorphisms in IL-2, IL-4, IL-6 and IL-10 genes have shown association with clinical disease, and the discriminative value of IL-4 to differentiate latent versus clinical RHD has been demonstrated [99]. Additionally, interleukins IL-4, IL-8 and IL-1RA seem to predict progression from latent to clinical disease, while in individuals with advanced RHD co-regulated expression of IL-6 and TNF-α associate with severe valvular dysfunction, and higher IL-10 and IL-4 levels predicted adverse clinical outcomes [100].

The biggest advance in the field of RHD genetics has been genome-wide association studies (GWAS), of which four studies have been published to date (Table 1) [93,101,102,103]. GWAS are considered more suitable studies for complex diseases in which large numbers of variants can be tested by comparing single-nucleotide polymorphism (SNP) distributions in patients with the disease against selected controls [104].

These GWAS studies have thus far landed support for the presence of significant heritability in RHD, which is likely to be polygenic. The RhEumatiC Heart diseAse Genetics (RECHARGE) study is currently running in Rwanda, using next generation genetic sequencing on an approximate sample of 1000 participants, and is expected to be completed in 2024 (ClinicalTrials.gov Identifier: NCT02118818). The importance of investing resources in the genetic association of RHD lies in the potential to substantially contribute towards understanding disease pathogenesis and etiology and subsequently the prospect of contributing towards novel or repurposed therapeutics and vaccine development [92].

## 5. Advances in the RF/RHD Continuum

Even though all RHD is, in theory, preventable, advances in prevention of RHD in low-resource settings have been slow. Nevertheless, the past decade has seen a revived dedication and commitment to further understanding and best clinical practices at the primordial, primary, secondary and tertiary levels of prevention. The search for a safe and effective GAS vaccine continues, and this will likely form the monumental advance that will deal the final blow to RF/RHD in all regions of the world, regardless of prevailing socioeconomic and health system determinants.

### 5.1. Advances in Primordial Prevention

Recent studies have aimed to provide a better description of the influence of socioeconomic determinants, health hardware, and transmission dynamics of GAS. Household crowding has been shown to have the strongest evidence of causal association with GAS infection and ARF/ RHD, increasing the risk by 1.7- to 2.8-fold. Dwelling characteristics such as construction type, house condition, dampness, and ventilation are also associated with an increased risk of infection that ranges from 1.8 to 3.6-fold [105]. The greater risk of having RHD among first degree relatives of individuals with clinical RHD may be secondary to the shared socioeconomic conditions [106,107]. Approaches targeting these primordial determinants have been designed and are being implemented via a community approach in Australia [108].

#### Group A Streptococcus Vaccines

Efforts to create a vaccine to prevent GAS infections have been ongoing since 1923. However, the first vaccines were ineffective and highly reactogenic, raising concerns, though likely unfounded or overblown, about the potential for vaccines to increase ARF predisposition; this resulted in the US FDA stopping GAS vaccine trials in 1970s for over 30 years [109,110]. In the past two decades, studies on vaccine development have improved with advances in genomics, proteomics, and immunomics; still, most vaccines are in pre-clinical testing, and few have reached phase I and II trials. Currently, there are no licensed vaccines available [110].

Vaccine candidates include multivalent M protein-based vaccines, M protein vaccines containing conserved C-repeat epitopes, cell wall carbohydrate vaccines, and non-M protein multi-component vaccines. Table 2 provides a summary of vaccines that are in the development pipeline. Vaccines based on group A carbohydrate (GAC), a polysaccharide present in *Streptococcus pyogenes* cell wall, have not shown expressive results yet [110].

Prevailing challenges in vaccine development are multifactorial, ranging from incomplete understanding of the basic science of GAS/RF/RHD to lack of commercial stakeholder interest. Specific challenges for GAS vaccine development include: (a) extensive genomic heterogenicity of Strep A and subsequent protein sequence variations, limiting the effectiveness of the vaccine over different populations [110]; (b) complexity of global GAS epidemiology; (c) incomplete understanding of ARF pathogenesis; (d) risk of serious autoimmune reactions to vaccines [109]; (e) dependence on controlled human infection models for vaccine development as GAS is strictly a human pathogen, thus precluding the use of animal models [111]; (f) lack of consensus on clinical endpoints for establishment of proof of concept, (g) limited market in high-income countries; and h) lack of commercial interest [110].

In order to overcome the myriad of challenges in GAS vaccine development, in 2018 the World Health Assembly (WHA) launched a Global Resolution calling for better control and prevention of GAS infections and RHD [112]. In 2019, the Strep A Vaccine Consortium (SAVAC) was formed to work along with WHO [110]. Furthermore, through the Coalition to Advance Vaccines Against Group A Streptococcus (CANVAS), the Australian and New Zealand governments have designated significant funding to support the development of a vaccine against GAS pharyngitis [113]. If the current momentum in vaccine development is maintained, it would be reasonable for research teams to start mapping out and developing potential GAS vaccine trial sites in different RHD endemic regions and countries.

### 5.2. Advances in Primary Prevention

#### 5.2.1. Diagnosis of Group A Strep Pharyngitis

Microbiological culture of a throat swab remains the gold standard for diagnosing GAS pharyngitis despite the prevailing limitations of prohibitive cost at a population level, the long turn-around time precluding confirmed diagnosis in a single clinic visit, and lack of readily available culture-able laboratories in LMICs. Recognizing these limitations, alternative diagnostic tests which are less resource intensive have been developed and continue to evolve.

Clinical decision rules can obviate the need for expensive bacteriological diagnostic tools since they do not require specialized equipment and are easy for providers to implement [119,120,121]. There has been a proliferation of Clinical Decision Rules (CDRs) for diagnosis of GAS pharyngitis in recent years [119,120,122,123,124,125,126,127]. Recent developments relevant to RHD endemic regions include: (a) validation studies of existing CDRs in RHD endemic regions [128] with the conclusion that diagnostic performance varies considerably in different regions of the world, thus highlighting the importance of evaluating and validating CDRs in local settings before they are rolled out as the standard of care; and, (b) development of CDRs in RHD endemic regions, such as the Cape Town Clinical Decision Rule [119] in South Africa. The Cape Town Clinical Decision Rule is likely a more relevant application to similar settings in sub-Saharan Africa. Tailoring decision rules to a specific population is a critical research area for population specific investment in LMICs.

Rapid Antigen Detection Tests (RADTs) have been in clinical use for four decades, with attractive features such as a quick turn-around time (<10 min), low cost, and ease of use. Recently, we have learnt that some external performance factors, such as inadequate staff training, substantially reduced the accuracy of these tests [129]. There is also greater understanding of factors that potentially increase heterogeneity for tests that have similar sensitivity and specificity [130], such as differences in throat culture sample collection [130,131], experience of the person performing the RADT, absence of a universally accepted blood agar plate culture method to serve as a reference standard [132], patient-level characteristics like clinical presentation and inoculum size [133], and spectrum bias [134]. However, despite multiple advantages, RADTs remain vastly unavailable and are considered expensive for many low resource settings. There is an overarching need to make existing RADTs available and affordable for use in low resource settings.

More recent progress in the area of point-of-care tests for GAS pharyngitis include Nucleic Acid Amplification Tests (NAATs), which have much better sensitivity and specificity than RADTs [135]. For example, the sensitivity and specificity of the Illumigene assay are estimated to be in excess of 99% [135,136,137]. This high performance, coupled with speed of results, makes NAATs ideal candidates for point-of-care use in the clinical environment. Several NAATs have received approval over the past six years [138], but their high cost has precluded widespread use. Despite the prevailing cost, NAATs are being increasingly investigated as low-cost, integrated tools for use in low resource settings [139,140]. Research should strive towards the development of molecular diagnostic tests using a “pharyngitis panel” of targets.

Electrochemical detection which uses DNA has been proposed as an affordable, effective method for diagnosing GAS; results are available in 30 min and 100% specificity has been reported [141,142].

Machine learning and artificial intelligence techniques are in development to aid the diagnosis of strep throat through throat image processing [143] and automated examination of throat cultures to identify GAS [144]. Neural networks have also been suggested to assist diagnosis, with reported correct diagnosis of pharyngitis in 95.4% of cases [145]. The Strepic^®^ device, a qualitative point-of-care clinical prototype, has been designed specifically as a viable, low-cost, commercially realizable autofluorescence-based diagnostic test (ClinicalTrials.gov Identifier: NCT03777098).

#### 5.2.2. Treatment of Group A Strep Pharyngitis

Penicillin remains the first line recommendation for the treatment of GAS pharyngitis. Two Cochrane reviews have found no evidence to change this recommendation [146,147]. Monthly intramuscular injections of benzathine benzyl penicillin (BPG) also remain the gold standard for secondary antibiotic prophylaxis for the prevention of RHD [148,149]. Recent strategies to diminish pain [150] associated with BPG administration include the addition of lignocaine to BPG solution [151,152] and the use of pain distraction methods (buzzy R) [152,153,154,155]. There is also work under way in developing implantable and longer acting BPG delivery devices [156,157,158]. The WHO added BPG to the essential medicines list for member states as a way to increase access, however, disruptions in global and local BPG supply chains are not uncommon [159,160]. Additionally, on a threatening note, a recent report described geographically widespread reduced in-vitro susceptibility of *Streptococcus pyogenes* to beta-lactam antibiotics associated with mutations in the pbp2× gene [161], warranting enhanced surveillance and further epidemiological and molecular genetic study of this potential emergent antimicrobial problem.

Community and provider knowledge and awareness remain a pillar of primary prevention strategies. While the previous campaigns heavily utilized print and mass media [36,162,163], increasing technologies, access to electronic devices and internet usage (including in low resource settings), have resulted in the employment of electronic avenues of education [164]. The COVID-19 pandemic has exerted further pressure for change in communication strategies.

Streptococcal carriage rates often vary between communities and by season, especially in endemic countries [165,166]. It has been postulated that the GAS carrier state is not implicated in the pathogenesis of RF/RHD and that transmission of GAS is almost limited exclusively to individuals with acute GAS infection [167,168]. Therefore, antibiotic treatment for eradication of GAS is only recommended for individuals with acute GAS infection. However, this traditional benign dogma of the carrier state is being challenged and re-visited. A GAS carriage paradigm seeks to gain a better understanding of the role of GAS carriage in the pathogenesis of RF and RHD.

### 5.3. Advances in Secondary Prevention

The Jones criteria for the diagnosis for ARF were first established in 1944 [169]. Since their inception, the criteria have undergone multiple revisions and updates, most recently in 2015 [7]. This most recent revision addressed two significant features: (a) distinguished criteria between low-risk and moderate-to-high risk populations based on ARF incidence or RHD prevalence; and (b) recommending echocardiography of all suspected cases of ARF and incorporating subclinical carditis as evidenced by echocardiography as a major criteria. Differentiating criteria for low-risk and moderate-to-high risk populations aimed to increase sensitivity in endemic regions while retaining specificity in low-risk areas, thus making the criteria more globally relevant. However, ARF still remains a clinical diagnosis with no single confirmatory test, and there is ongoing work attempting to identify a unique immune signature that could be used to reliably diagnose ARF [170].

In routine clinical practice it is often not feasible to obtain both acute and convalescent sera and therefore, the absolute quantitative measure of anti-streptolysin (ASO) titers is used for diagnostic value [171]. However, there is wide geographic variability of the 80th percentile upper limit of normal (ULN) cutoffs for ASO titers [172,173]. It is therefore important to establish ULN ranges of ASO titers for various age groups in different geographic locations. Accordingly, there have been recent studies in some RHD endemic regions describing their population-specific streptococcal antibody titers [174].

As previously mentioned, there is recent evidence for the benefit of secondary antibiotic prophylaxis in latent RHD [53]. The GOAL trial showed that among children and adolescents 5 to 17 years of age with latent RHD, secondary antibiotic prophylaxis reduced the risk of disease progression at 2 years [53]. This new evidence provides added justification for echocardiographic screening and active case detection of RHD as a viable approach for the control of RHD.

### 5.4. Advances in Tertiary Prevention

#### 5.4.1. Valve Surgery vs. Valve Repair

Mitral valve (MV) repair, rather than MV replacement, is currently the preferred surgical management for patients with mitral regurgitation (MR) as evidence suggests better short and long-term outcomes [175,176,177,178], including shorter hospital stays and fewer infections [179]. However, MV repair requires refined surgical skills with an associated learning curve, adequate surgical environment, and echocardiographic guidance by an expert, in addition to relatively favorable valve anatomy. For these reasons, in low-income areas where RHD is prevalent, surgical teams often have more expertise in performing MV replacement, making it the first choice for surgical intervention [178] despite recent reports demonstrating higher survival rates in patients undergoing MV repair vs MV replacement.

#### 5.4.2. From Valvotomy to Balloon Mitral Valvuloplasty (BMV)

There is a growing body of evidence about BMV for RHD, and the indications have considerably expanded in the past decade to include challenging and unfavorable MV involvement. Furthermore, additional prognostic parameters have been investigated—such as asymmetrical commissural fusion and atrioventricular compliance—and the Wilkins score was updated with parameters derived from international RHD cohorts [180]. Meanwhile, percutaneous valve interventions continue to evolve worldwide, with transcatheter aortic valve implantation for the treatment of aortic stenosis being a prime example [181]. With its successful implementation, percutaneous valve intervention is gaining attention as a feasible treatment for aortic and mitral disease, as well as an option for repeated interventions—common in RHD—and should be established in the upcoming decades [182,183].

#### 5.4.3. Transcatheter Aortic Valve Replacement (TAVR) for Rheumatic Aortic Stenosis

TAVR is an established minimally invasive alternative to surgical aortic valve replacement (SAVR) in patients with severe aortic stenosis (AS) and calcific aortic valve disease [184,185]. In RHD with aortic involvement, SAVR is still the first choice due to the lower degree of valve calcification in most cases [185,186], younger age of patients, and limited scientific evidence of TAVR [186]. Until recently, knowledge surrounding TAVR’s application for RHD patients was limited to case series and reports [185,187,188,189]. However, there is contemporary evidence for non-inferiority of TAVR to SAVR in rheumatic AS, safety of TAVR, and short and intermediate term outcomes of TAVR [185,188]. These novel data have changed the perspective of TAVR for RHD, suggesting TAVR as a feasible approach for RHD patients with predominant aortic valve involvement. However, wider spectrum studies are warranted in order to achieve generalization of findings.

#### 5.4.4. Medical Management of Clinical RHD

Patients with symptomatic RHD are a burden of low-income regions where surgical treatment options are limited. As medical therapy has yet to slow the progression of the disease, treatment targets symptom relief by addressing underlying left ventricular dysfunction and heart failure [178]. There have been no significant developments to advance medical management strategies.

Atrial fibrillation (AF) is a major cause of morbidity and mortality in patients with RHD [62,178,190]. Anticoagulation is recommended to reduce the risk of cardioembolic events in patients with AF [62,190,191]. Oral Vitamin K antagonists are the recommended drugs of choice [178], but they are associated with multiple challenges both for health care providers and patients [190,191]. New evidence may revolutionize anticoagulation in rheumatic atrial fibrillation, as novel oral anticoagulants (NOACs)—not yet formally recommended for AF in the presence of RHD—have proven to be non-inferior to warfarin in a Brazilian trial which allowed the inclusion of RHD patients with atrial fibrillation and bioprosthetic mitral valves [192]. The multicenter INVICTUS trial [193] for warfarin versus rivaroxaban in rheumatic AF is nearing completion and may bring more definite conclusions in the near future. However, even if NOACs become a standard for anticoagulation in RHD, access and cost-effectiveness will require careful discussion for their use in LMICs [194].

Given the lack of scientific evidence in the reduction of infective endocarditis (IE) burden in the absence of prosthetic valves or previous history of IE, new guidelines indicate a limited role of antibiotic prophylaxis for dental procedures [195]. Today, recommendation for prophylaxis is limited to high-risk patients—noticeably those following valve replacement or with a previous IE episode—undergoing high-risk procedures with potential bacterial translocation [196]. Good dental hygiene and regular dental cleanings likely play an important role in IE prevention and should be emphasized [195].

## 6. Rheumatic Heart Disease in Pregnancy

Preexisting cardiac disease is a major contributor to maternal mortality worldwide, especially in LMICs [197]. Recognition of RHD in pregnancy is extremely important, especially given the higher prevalence of RHD in women, younger age of patients with RHD in LMICs [190], and the significant risk posed by RHD during pregnancy. Recent echocardiographic screening studies in sub-populations of pregnant women in RHD endemic regions have provided epidemiologic descriptions of the burden and outcomes of RHD in pregnancy [197,198], reinforcing the need for programs dedicated to early diagnosis and prioritization of cardiovascular care during family planning and pregnancy in RHD-endemic regions. Family planning with adequate preconception counseling should be provided to known RHD patients. However, that is often not the reality. For example, only 5% of women with prosthetic valves and 2% of women with severe mitral stenosis in the REMEDY study were on contraception [178] Finally, in pregnant women with primary indications for anticoagulation, particularly mechanical valves, management remains complex and challenging [178,190]. The ideal choice of bioprosthetic valves in women of child bearing potential to circumvent the need for anticoagulation is still not feasible in many RHD endemic regions because of the need for re-operation.

## 7. Other Recent Advances

### 7.1. Understanding of RHD-HIV Co-Infection

The regions of the world with the highest RF and RHD prevalence are also those with the greatest human immunodeficiency virus (HIV) prevalence, but there has been little investigation into the epidemiology of the dual burden of these diseases [58,199]. Previous small cross-sectional surveys in Uganda have reported conflicting results regarding the association between HIV and RHD [200,201,202]. A recent description of the epidemiologic profile and longitudinal outcomes of an HIV-RHD comorbid population, however, revealed that the prevalence of HIV in this cohort was not significantly higher than that of the general Ugandan population, with rates of 3.6%, compared to a national range of 2.1% in children and 6.2% in people over 15 years of age [203]. Furthermore, comorbid individuals did not appear to suffer higher mortality rates than those with RHD alone. However, HIV-RHD comorbid subjects had nearly threefold higher odds of having suffered a stroke or transient ischemic attack compared to those without HIV [203,204]. Overall, these data imply that aside from elevated cerebrovascular accident risk, RHD defines the short-term clinical outcomes of this group of people more than HIV does [205]. As major innovations in HIV treatment have significantly improved the long-term survival of people living with HIV, comorbid noncommunicable diseases have begun to dominate their morbidity and mortality [206]. This presents an opportunity for RHD care networks to leverage and adopt tools and innovations developed for the HIV public health effort, such as the Cascade of Care, to benefit not just HIV-RHD comorbid patients, but also individuals residing in LMICs affected by RHD in general [207,208].

### 7.2. Understanding of RHD-Associated Costs in Endemic Regions

RHD costing data from endemic countries and regions has been sparse through the years. However, incremental cost-effectiveness ratios (ICERs) have been recently calculated with comparison of primary, secondary, and tertiary prevention in the African setting [209]. Findings revealed that scaling up primary prevention would be a cost saving approach, with a negative ICER (USD −2539 per DALY averted), whilst putting efforts into secondary prevention programs would be cost-effective (USD 752 per DALY averted) [209]. The investments required for local surgery capacity were high with limited impact (ICER of USD 23,827 per DALY averted for building a local center) [209]. Several other publications have separately explored optimum cost-effective strategies for primary prevention [210], secondary prevention [211,212,213,214], and a combination of both [214,215,216]. A most recent modeling study for the prevention and management of RHD in the African Union concluded that, “In the short term, costs of secondary prevention and secondary and tertiary care for RHD are lower than for primary prevention, and benefits accrue earlier [217]”. Most analyses make use of Markov models, and similar challenges relating to lack of precise transition probabilities on which to base the calculations have been reported [209,212,215].

Additionally, recent studies have described the near-atrocious economic impact of RHD at the household, health system, and national level in endemic countries such as Uganda [218], South Africa [219,220], India [221,222], and others [223].

### 7.3. Global Efforts, Advocacy, and Stakeholder Engagement in the Fight against RHD

The growing research and healthcare/awareness projects in RHD worldwide, combined with a considerable boost in research interest over the past two decades, has raised multisectoral attention on this neglected disease. This has resulted in international coordination of efforts which have been multifaceted, with ambitious aims (Table 3).

As an example of practical outcomes of this international coordination, in 2011 the United Nations set key targets to reach by 2025, including a reduction in the risk of premature noncommunicable disease death–markedly cardiovascular disease–by 25% by 2025; RHD was inclusive in this agenda [45], premiering its position in a broad recommendation of development goals. Consequently, multiple authoritative individuals [42,43], societies, task forces, unions and federations, at regional, continental and international levels have published position statements [44,45,46,226,227], serving as a basis for guiding research and healthcare initiatives, helping define “next steps” and priorities for the world scientific agenda, and definition of priority actions. The WHF has played a particularly pivotal role, with its first broad position statement on RHD released in 2013 reinforcing key aims and actions in a similar direction of African statements [227]. More recently, there was a re-dedication of the American Heart Association (AHA) on the RHD agenda [228] with resumption of its intensive work on the development of guidelines and statements [7,178], research funding through its councils, and a working group dedicated to RF and RHD.

These aforementioned position papers and statements from different organizations and regions, and the intensive collaborative multisectoral efforts worldwide, were crowned by the 2018 World Health Assembly Global Resolution to end RHD [47,224,225]. The key charges that this resolution brings to member states are listed in Table 3. Following this unprecedented and supportive resolution, health systems should move towards its recommendations, increasing investment in primary care and infrastructure, sanitation and housing, medical supply and building capacity for RHD prevention and management, including advanced resources.

## 8. Conclusions

In conclusion, we have entered a period in history that could again be marked by rapid progress towards elimination of RHD, this time on the global stage. Application of modern scientific techniques for improving RHD prevention, diagnosis and treatment have the potential to revolutionize our approach to RHD and are urgently needed. Progress is being made in basic science, clinical, translational, and population-based research. However, to sustain and accelerate this trajectory, substantial multisector investment, including research funding, capacity building, and resources for education and awareness are urgently needed. The solutions are within our grasp and with adequate investment, we can substantially reduce the global burden of RHD in our lifetime.

## Figures and Tables

**Figure 1 pathogens-11-00179-f001:**
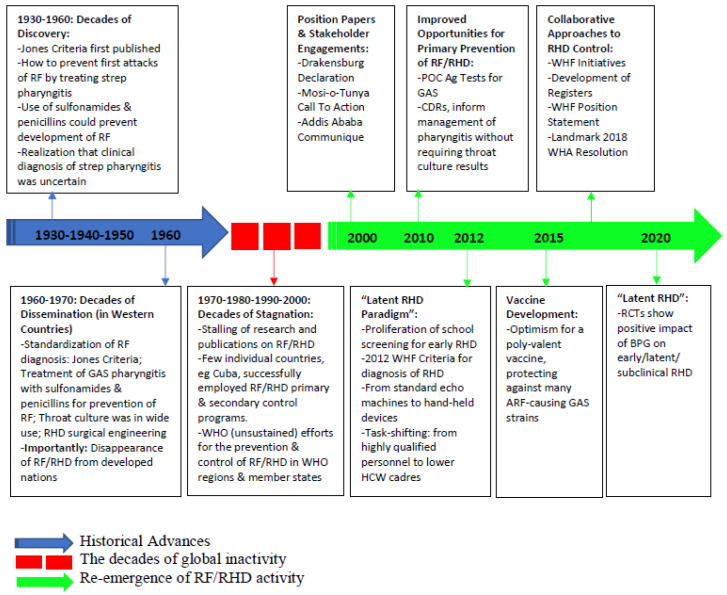
Central Illustration—timeline of advances in RF/RHD. Abbreviations: GAS, group A *Streptococcus*; RF, rheumatic fever; ARF, acute rheumatic fever; RHD, rheumatic heart disease; POC, point of care; Ag, antigen; CDR, clinical decision rule; BPG, benzyl benzathine penicillin; HCW, health care worker; RCT, randomized clinical trial; WHA, World Health Assembly; WHF, World Heart Federation; WHO, World Health Organization.

**Table 1 pathogens-11-00179-t001:** Findings from genome-wide association studies. Abbreviations: GAS, group A *Streptococcus*; HLA, human leukocyte antigen; IGH, immunoglobin heavy chain; GWAS, genome-wide association studies; GWS, genome-wide significance.

Author	Countries Involved	Sample	Platform	Key Findings
Auckland et al., 2020 [93]	Fiji, India, UK	822 cases; 1800 controls	HumanCore- 24 BeadChip (Illumina Inc., San Diego, CA, USA) UK Biobank Axiom Array (Affymetrix, USA)	HLA class III *PBX2* region rs201026476, chromosome 14 (OR 1.99, 95% CI 1.58–2.51, *p* = 7.45 × 10^−9^) reached GWS Independent association was maintained in HLA class III region after conditioning and results replicated by validating through UK dataset.
Parks et al., 2017 [101]	Eight Oceanian countries	1006 cases; 1846 controls	Low-density 300 k Illumina HumanCore platform	IGHV4-61 02 rs11846409 on chromosome 14 (1.4× risk) (OR 1.43, 95% CI 1.27–1.61, *p* = 4.1 × 10^−9^) that reached GWS
Gray et al., 2017 [102]	Northern territory of Australia	398 cases; 865 controls	HumanCore-24 Bead Chip (Illumina Inc., San Diego, CA, USA)	HLA-DQA1 rs9272622 on chromosome 6 (protective) (OR = 0.90, *p* = 1.86 × 10^−7^) did not reach GWS
Machipisa et al., 2021 [103]	8 African countries	2548 cases; 2261 controls Family group -348 participants (118 trio families)	Infinium Human Omni 2.5–8 (Illumina Inc., San Diego, CA, USA)	GWS association at 11q24.1 (rs1219406); (OR 1.65; 95%CI, 1.48– 1.82; *p* = 4.36 × 10^−8^) for black African individuals but not admixed African individuals or other external datasets Replicated rs11846409 IGH locus GWAS [101] in admixed African individuals

**Table 2 pathogens-11-00179-t002:** Group A *Streptococcus* vaccines that are in the development pipeline. * Pioneered by University of Tennessee and Dalhousie, Canada; ** Pioneered by Queensland Institute of Medical Research, Australia; Pioneered by University of São Paulo, Brazil.

Type of Vaccine	Stage of Development
StreptAvax: 26-valent vaccine	Phase I and II trials demonstrated good safety, tolerance and immunogenicity [114,115]; however further studies stopped for commercial reasons
StreptAnova *: 4 recombinant proteins	The 4 recombinant proteins represent 30 different M-types prevalent in North America and Europe Phase I trial: demonstrated good tolerance and immunogenicity in adults [116].
MJ8VAX **: based on C-terminus of the M protein	Phase I trial: demonstrated that a single intramuscular dose of the vaccine was safe, well tolerated and immunogenic, but anti-J8 IgG concentration decreased after 180 days post immunization [117].
StreptInCor: peptide vaccine containing T and B cell epitopes of the M protein CRR	Good results in models [118].
Multi-component vaccines [110]	3-Combo: SpyCEP, SpyAD, SLO; provides protection in models 5-Combo: ADI, TF, C5a peptidase, SpyCEP and SLO 5-CP: demonstrated protection against intranasal, skin and systemic challenges of GAS Spy-7: showed significant reduction in the dissemination of types M1 and M3 GAS

**Table 3 pathogens-11-00179-t003:** Summarized aims of global efforts in the fight against RHD, and key charges from the 2018 WHA resolution to member states. Abbreviations: RHD, rheumatic heart disease; WHO, World Health Organization; WHF, World Heart Federation; WHA, World Health Assembly.

(Ambitious) Aims of Global Efforts in the Fight against RHD
• Developing a collaborative agenda to address the key health impacts of RHD worldwide (premature morbidity and mortality, maternal deaths) • Improving access to basic healthcare, from prophylaxis to more advanced therapies • Sensitizing stakeholders and the political sector on the importance of RHD and the need for reducing its burden, especially in the hardest hit regions • Developing position statements, guidelines, and calls-to-action focused on RHD • Including RHD as a priority in the world agenda, establishing action plans and goals led by international organizations such as the WHO, WHF, and key cardiovascular societies
**Key Charges from the 2018 WHA Resolution to Member States** [47,224,225]
• Accelerate multisectoral efforts to improve socioeconomic determinants of RHD • Estimate the burden of disease and implement multisectoral RHD programs • Improve access to primary health care, including RHD prevention and control • Ensure access to cost-effective essential laboratory technologies and medicines for RHD • Strengthen national and international cooperation to address RHD.

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
