# Peer review of "Recent Advances in the Rheumatic Fever and Rheumatic Heart Disease Continuum"

_pathogens, 2022, doi:10.3390/pathogens11020179_

Round 1
Reviewer 1 Report
This is a very comprehensive review on rheumatic fever and rheumatic heart disease, an often neglected disorder. The manuscript is very well written and easy to read. I particularly enjoyed the tables which provide a very nice, updated summary of the evidence on a variety of relevant topics.
My only comment/suggestion is to modify the Abstract to make it less "introductory" and instead, provide a summary of the various topics or recent advances as detailed in the text.
Reviewer 2 Report
Authors in their narrative review article titled “ Recent Advances in the Rheumatic Fever and Rheumatic Heart 2 Disease Continuum” highlighted the timelines of advances in RF/RHD into three phases namely; historical advances(1930-1960s), decades of global inactivity(1970-200) and reemergence of RF/RHD activities (2000-2020). Overall manuscript is well structured and well drafted. The advancement taken place before 1960s is well covered.
- The time line between 1970 to 2000 described as phase of inactivity probably is not correct.
- Number of survey studies estimating the prevalence of RF/RHD mostly in school children has been reported from south Asian countries.
- The manuscript has been drafted mostly in the context of Sub Saharan Africa.
- Data from India reveal there has been declining trends in the burden of RF/RHD after 2000 compared to before 2000 using similar method of case detection.
- Thus in the section of Advances in Understanding the Global Burden of RHD; global trends in the burden of RF/RHD should be described based on population based survey studies using similar methods of detection at different timelines available in different regions of the globe
- Advancements in the understanding of impact of RF/RHD on public health have not been adequately highlighted and covered in the manuscript under heading RHD Registries and Big Data.
- It would be desirable to restructure the draft giving due share to this important facet.
- There are publications from northern India, Northern Territory of Australia about incidence of; mortality, stroke, heart failure hospitalization, atrial fibrillation and their risk determinants, in patients with RF/RHD in recent past based on large hospital based longitudinal studies.
- Gender based differences in the prevalence, nature of valvular dysfunction should also be described in the manuscript along with possible reasons as no gender based differences in the incidence of RF is reported.
- Potential areas of research in the basic sciences for future trying to understand the pathogenic pathways responsible for progression and lading to obstructive, incompetent and combination of valvular dysfunction
Author Response
Please see the attached response.
